# Sliced Cramér Synaptic Consolidation for Preserving Deeply Learned Representations

**Soheil Kolouri, Nicholas A. Ketz, & Praveen K. Pilly**
HRL Laboratories, LLC
Malibu, CA, 91301, USA
{skolouri, naketz, pkpilly}@hrl.com

**Andrea Soltoggio**
School of Computer Science,
Loughborough University,
Leicestershire, UK
a.soltoggio@lboro.ac.uk

## Abstract

Deep neural networks suffer from the inability to preserve the learned data representation (i.e., catastrophic forgetting) in domains where the input data distribution is non-stationary, and it changes during training. Various selective synaptic plasticity approaches have been recently proposed to preserve network parameters, which are crucial for previously learned tasks while learning new tasks. We explore such selective synaptic plasticity approaches through a unifying lens of memory replay and show the close relationship between methods like Elastic Weight Consolidation (EWC) and Memory-Aware-Synapses (MAS). We then propose a fundamentally different class of preservation methods that aim at preserving the distribution of the network's output at an arbitrary layer for previous tasks while learning a new one. We propose the sliced Cramér distance as a suitable choice for such preservation and evaluate our Sliced Cramér Preservation (SCP) algorithm through extensive empirical investigations on various network architectures in both supervised and unsupervised learning settings. We show that SCP consistently utilizes the learning capacity of the network better than online-EWC and MAS methods on various incremental learning tasks.

## 1 Introduction

Incremental learning without catastrophic forgetting is one of the core characteristics of a lifelong learning machine (L2M) and has recently gained renewed attention from the machine learning community. In real-world applications, the input distribution of the data (e.g., sensory inputs) is prone to constant changes due to environmental variations (e.g., seasonal changes), exposure to new situations (e.g., change in the surface friction), sensory malfunction (e.g., water droplets on a camera), among others. It is therefore desirable to continue to train the base computational model only on the new data/task and incrementally accumulate knowledge to improve the performance of the system over time, as opposed to retraining the model on the composition of old and new data.

The existing computational models, for instance deep convolutional neural networks (CNNs), face two fundamental issues regarding incremental learning, 1) *catastrophic forgetting* (McCloskey & Cohen, 1989), which refers to the forgetting of previously acquired knowledge when learning new tasks as a result of interference between the old and new tasks, and 2) *intransigence*, which refers to the inability to acquire new knowledge while trying to preserve old knowledge (e.g., reducing the learning rate) Chaudhry et al. (2018). Note that we use the term 'knowledge' here to indicate the input/output behavior of the computational model as in (Hinton et al., 2015). A successful incremental learner should be able to overcome both forgetting and intransigence. The commonly used strategies to overcome catastrophic forgetting include:

1. selective synaptic plasticity to preserve learned knowledge (Kirkpatrick et al., 2017; Zenke et al., 2017; Aljundi et al., 2018; Chaudhry et al., 2018), which is rooted in the idea of homeostatic plasticity in neuroscience,

2. additional neural resource allocation to learn new knowledge and preserve old knowledge, (Rusu et al., 2016; Lee et al., 2017; Li & Hoiem, 2017; Rannen et al., 2017; Schwarz et al., 2018; Li et al., 2019), which is similar to neurogenesis in the hippocampus,

3. memory and experience replay (Rebuffi et al., 2017; Shin et al., 2017; Wu et al., 2018; Hu et al., 2019; Rostami et al., 2019), which is based on the well-established theory of complementary learning system (CLS) (McClelland et al., 1995).

Each framework has its advantages and disadvantages, and Parisi et al. (2019) provide an excellent survey of these methods. We note that the term "synaptic weights" refers to the strength of a connection between two nodes. The term 'plasticity' is used analogously to 'neural plasticity' in the human brain, which refers to the ability of the neurons to change their synaptic weights. The term "selective plasticity" refers to the desired capability of a network to selectively increase or decrease the plasticity of individual synapses throughout the neural architecture.

Our focus in this paper is on selective synaptic plasticity. The standard deep neural network architectures are uniformly plastic; hence, all neurons are prone to changes during training, and this powerful capability is also the demise of these networks and leads to catastrophic forgetting. The idea of selective synaptic plasticity is to partially preserve synapses that are critical for previously learned tasks by rigidifying those synapses (i.e., to enforce critical synapses to change less). Rigidifying the network over time leads to a loss of learning capability for future tasks, which is known as 'intransigence' in the literature. Selective synaptic plasticity, by itself, could not fully overcome intransigence. A combination of strategies like efficient memory replay for reconsolidation, neurogenesis, and selective synaptic plasticity could lead to superior methods that defeat both catastrophic forgetting and intransigence. Chaudhry et al. (2018), for instance, provide such a combination of memory replay and selective synaptic plasticity. Also, there have been various efforts toward making the idea of neural resource allocation scalable, the progress and compress work Schwarz et al. (2018), and the incremental moment matching Lee et al. (2017) work fall under this category.

In this paper, we focus on selective synaptic plasticity to preserve learned representations in a deep neural network. Inspired by (Chaudhry et al., 2018), we take a geometric view and devise a new method for selective synaptic plasticity. The proposed method is fundamentally different from the previous approaches like Elastic Weight Consolidation (EWC) (Kirkpatrick et al., 2017) and Memory Aware Synapses (MAS) (Aljundi et al., 2018), which we indicate as sample-based approaches. Instead, we focus on identifying synaptic importance parameters that preserve the '*distribution*' of the latent representation of a task. Focusing on preserving the distribution of the latent representation of a neural network at an arbitrary layer, as opposed to the expected change in network's response for individual samples, enforces a less restrict regularization on the network, and enables a better utilization of the network learning capacity. The primary concept of sample-based versus distribution-based regularizations are visualized in Figure 1. Similar to the MAS framework, our proposed method denoted as Sliced Cramér Preservation (SCP) is also able to preserve a task representation in any layer of a neural network, hence, enabling its application to various unsupervised or self-supervised learning settings.

Our specific contributions in this paper are:

- Introducing a distribution-based regularization using the sliced Cramér distance (aka, Cramér Wold distance (Tabor et al., 2018)) for selective synaptic plasticity that preserves the distributions of the representations of previously seen tasks at an arbitrary layer of a deep neural network while learning a new task.

- Providing a geometric interpretation for the MAS algorithm (Aljundi et al., 2018) that further gives insight into the otherwise heuristic choices made in the approach.

- Comparing the proposed method to online-EWC and MAS on the benchmark permuted MNIST dataset, sequential unsupervised learning with auto-encoders, and the more interesting problem of semantic segmentation of driving scenes, and demonstrating significant improvements in overcoming catastrophic forgetting and intransigence over these methods.

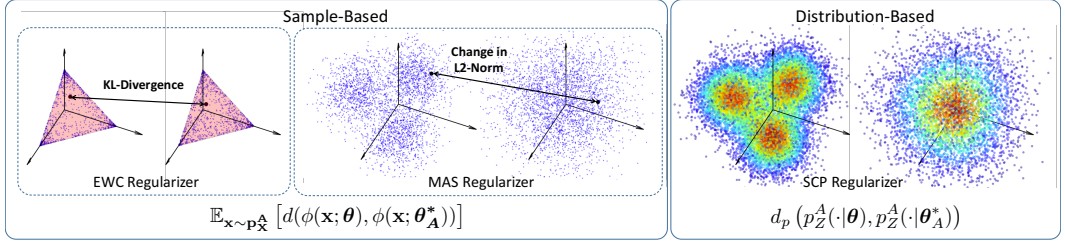

Figure 1: Sample-based approaches regularize the learning by the expected change, as measured by a dissimilarity measure, of the response of the network for individual samples from Task A, after learning Task A and during learning Task B, $\mathbb{E}_{\boldsymbol{x} \sim p_X^A}\left[d(\phi(\boldsymbol{x}; \boldsymbol{\theta}), \phi(\boldsymbol{x}; \boldsymbol{\theta}_A^*))\right]$, where $d(\cdot, \cdot)$ is a dissimilarity measure between two $K$-dimensional vectors. Therefore, the regularization is an empirical expected change of the response for samples. EWC and MAS fall under the sample-based category. The proposed distribution-based approach, on the other hand, regularizes the change in the overall distribution of the network's output for input samples from Task A, after learning Task A and during learning Task B, $d_p\left(p_Z^A(\cdot|\boldsymbol{\theta}), p_Z^A(\cdot|\boldsymbol{\theta}_A^*)\right)$, where $p_Z^A(\cdot|\boldsymbol{\theta})$ is defined in equation 8 and $d_p(\cdot, \cdot)$ is a distance measure between two probability distributions defined on $\mathcal{Z} \subseteq \mathbb{R}^K$.

## 2 PROBLEM SET-UP AND PRELIMINARIES

Consider data from a stream of tasks $\mathcal{X}^t = \{\boldsymbol{x}_i^t \sim p_X^t\}_{i=1}^{n_t}$, where $p_X^t$ is the probability density function (PDF) for task $t$ defined on $\mathbb{X} \subset \mathbb{R}^d$. We consider both supervised and unsupervised tasks, where in the supervised case the input sample, $\boldsymbol{x}_i^t$, is accompanied with the corresponding label $\boldsymbol{y}_i^t \in \mathbb{R}^K$. Let $\phi(\cdot; \boldsymbol{\theta}) : \mathbb{R}^d \to \mathbb{R}^K$ denote a parametric function (e.g., a neural network) that is to be optimized to solve the stream of tasks. In the supervised learning setting, we consider $\phi$ to be the mapping to the logits prior to applying the softmax layer.

### 2.1 A GEOMETRIC VIEW OF ELASTIC WEIGHT CONSOLIDATION

In their seminal work, Kirkpatrick et al. (2017) considered the problem of overcoming catastrophic forgetting in supervised and also reinforcement learning scenarios where a supervisory signal $\boldsymbol{y}$ exists, whether in the form of labels/annotations, or environmental rewards, respectively. Here we reiterate the geometric interpretation of the EWC framework following the work of (Chaudhry et al., 2018), which we will then adapt to define our generic consolidation framework. Let $p_{\boldsymbol{\theta}}(\boldsymbol{y}|\boldsymbol{x}) = \text{softmax}(\phi(x; \boldsymbol{\theta}))$, where $[p_{\boldsymbol{\theta}}(\boldsymbol{y}|\boldsymbol{x})]_j$ is the softmax probability of the j-th class. For simplicity, let us consider the case where we want to learn only two tasks consecutively, i.e., tasks 'A' and 'B.' Then, EWC ensures that while learning task 'B,' the conditional likelihood $p_{\boldsymbol{\theta}}(\boldsymbol{y}|\boldsymbol{x}^A)$ does not drift far from the optimal conditional likelihood $p_{\boldsymbol{\theta}_A^*}(\boldsymbol{y}|\boldsymbol{x}^A)$, where $\boldsymbol{\theta}_A^*$ are the parameters initially optimized for task A:

$$
\begin{aligned}
\arg\min_{\boldsymbol{\theta}} \tilde{\mathcal{L}}^B(\boldsymbol{\theta}) &= \arg\min_{\boldsymbol{\theta}} \mathcal{L}^B(\boldsymbol{\theta}) + \lambda \mathbb{E}_{\boldsymbol{x} \sim p_X^A}\left[D_{\mathrm{KL}}(p_{\boldsymbol{\theta}_A^*}(\boldsymbol{y}|\boldsymbol{x}) \,||\, p_{\boldsymbol{\theta}}(\boldsymbol{y}|\boldsymbol{x}))\right] \\
&= \arg\min_{\boldsymbol{\theta}} \mathcal{L}^B(\boldsymbol{\theta}) + \lambda \mathbb{E}_{\boldsymbol{x} \sim p_X^A}\left[\mathbb{E}_{y \sim p_{\boldsymbol{\theta}_A^*}}\left[log(\frac{p_{\boldsymbol{\theta}_A^*}(y|\boldsymbol{x})}{p_{\boldsymbol{\theta}}(y|\boldsymbol{x})})\right]\right]
\end{aligned} \quad (1)
$$

where $\lambda$ is the regularization coefficient. Note that equation 1 is essentially an optimization with a memory replay regularizer. Here, while learning task B, samples from task A (i.e., a memory buffer from this task) are fed through the network and the conditional likelihood is constantly checked against the optimal conditional likelihood for task A, $p_{\boldsymbol{\theta}_A^*}(y|\boldsymbol{x}^A)$, to ensure a minimal deviation from those parameters.

The key question answered by the EWC framework is on how to avoid memory replay and yet achieve a similar result, i.e., not forget the knowledge from old task (task A). The answer lies in the second-order Taylor expansion of the regularizer around the parameters optimized for the old task, $\boldsymbol{\theta}_A^*$. It is straightforward to show (Chaudhry et al., 2018) that the second-order Taylor expansion of

the regularizer is of the form:

$$\mathbb{E}_{\boldsymbol{x} \sim p_X^A} \left[ D_{\mathrm{KL}}(p_{\boldsymbol{\theta}}(\boldsymbol{y}|\boldsymbol{x}) \,||\, p_{\boldsymbol{\theta}+\delta\boldsymbol{\theta}}(\boldsymbol{y}|\boldsymbol{x})) \right] \approx \frac{1}{2} \delta\boldsymbol{\theta}^T F_{\boldsymbol{\theta}} \delta\boldsymbol{\theta} = \|\delta\boldsymbol{\theta}\|_{F_{\boldsymbol{\theta}}}^2 \tag{2}$$

where $F_{\boldsymbol{\theta}}$ is the Fisher Information Matrix (FIM) and is defined as:

$$F_{\boldsymbol{\theta}} = \mathbb{E}_{\boldsymbol{x} \sim p_X^A} \left[ \mathbb{E}_{y \sim p_{\boldsymbol{\theta}_A^*}} \left[ \left( \frac{\partial log(p_{\boldsymbol{\theta}}(y|\boldsymbol{x}))}{\partial \boldsymbol{\theta}} \right) \left( \frac{\partial log(p_{\boldsymbol{\theta}}(y|\boldsymbol{x}))}{\partial \boldsymbol{\theta}} \right)^T \right] \right] \tag{3}$$

see supplementary material for complete derivations. Therefore, when $\delta\boldsymbol{\theta} \to 0$ the KL-divergence regularizer enforces closeness of $\boldsymbol{\theta}$ to $\boldsymbol{\theta}_A^*$ in a Riemmanian pseudo-manifold induced by the FIM. Given that the number of parameters could easily reach several million in standard deep neural networks, it is practically infeasible to store and use the FIM matrix, $F_{\boldsymbol{\theta}}$. Therefore, Kirkpatrick et al. (2017) assume that $F_{\boldsymbol{\theta}}$ is diagonal and further approximate the KL-divergence with:

$$\mathbb{E}_{\boldsymbol{x} \sim p_X^A} \left[ D_{\mathrm{KL}}(p_{\boldsymbol{\theta}}(\boldsymbol{y}|\boldsymbol{x}) \,||\, p_{\boldsymbol{\theta}+\delta\boldsymbol{\theta}}(\boldsymbol{y}|\boldsymbol{x})) \right] \approx \frac{1}{2} \sum_{m=1}^{M} [F_{\boldsymbol{\theta}}]_{m,m} [\delta\boldsymbol{\theta}]_m^2 \tag{4}$$

where $M$ is the total number of parameters in the neural network. This leads to the main equation in the EWC framework:

$$\arg\min_{\boldsymbol{\theta}} \tilde{\mathcal{L}}^B(\boldsymbol{\theta}) \quad = \quad \arg\min_{\boldsymbol{\theta}} \mathcal{L}^B(\boldsymbol{\theta}) + \frac{\lambda}{2} \sum_{m=1}^{M} [F_{\boldsymbol{\theta}_A^*}]_{m,m} [\boldsymbol{\theta} - \boldsymbol{\theta}_A^*]_m^2 \tag{5}$$

From our point of view, the critical aspect of these derivations is the connection between memory replay and structural plasticity. Note that we started with equation 1, which uses the idea of a memory replay regularizer. Then by assuming $\delta\boldsymbol{\theta} \to 0$, using the second-order Taylor expansion of the KL-divergence around $\boldsymbol{\theta}_A^*$, and assuming that FIM is a diagonal matrix we arrived at equation 5, which provides the idea of synaptic importance parameters and is an embodiment of the selective synaptic plasticity framework. Next, we use this critical aspect and develop an analogous regularizer (i.e., based on memory replay) for the MAS algorithm.

## 2.2 Generalizing to Unsupervised Learning

The EWC framework as explained in the previous section preserves the softmax probability of samples from Task A while learning Task B. This limits the applicability of the method to networks with outputs living on a K-dimensional simplex, e.g., supervised learning and reinforcement learning where the network outputs a probability over a finite set of actions. More recently, Aljundi et al. (2018) presented their Memory-Aware-Synapses (MAS) framework, which lifts the requirement for EWC outputs to live on a simplex, and enables calculation of the synaptic importance parameters even in unsupervised learning and also during testing. While the method is exciting and practically very useful, there is no geometric motivation behind the algorithm. Here we reverse engineer the importance term used in MAS and show a simple regularizer that leads to the MAS algorithm and more importantly provides a geometric interpretation for the algorithm.

Let the regularizer for Task B, be the expected absolute difference between squared $\ell_2$ norms of the output of the network for samples from Task A, i.e.:

$$\arg\min_{\boldsymbol{\theta}} \tilde{\mathcal{L}}^B(\boldsymbol{\theta}) = \arg\min_{\boldsymbol{\theta}} \mathcal{L}^B(\boldsymbol{\theta}) + \lambda \mathbb{E}_{\boldsymbol{x} \sim p_X^A} \left[ \frac{1}{2} (\|\phi(\boldsymbol{x}; \boldsymbol{\theta})\|^2 - \|\phi(\boldsymbol{x}; \boldsymbol{\theta}_A^*)\|^2)^2 \right] \tag{6}$$

It is straightforward (see Supplementary material) to show that using the second-order Taylor expansion of the above regularizer leads to:

$$\arg\min_{\boldsymbol{\theta}} \tilde{\mathcal{L}}^B(\boldsymbol{\theta}) \quad = \quad \arg\min_{\boldsymbol{\theta}} \mathcal{L}^B(\boldsymbol{\theta}) + \lambda \sum_{m=1}^{M} [\Omega]_{m,m} [\boldsymbol{\theta} - \boldsymbol{\theta}_A^*]_m^2 \tag{7}$$

where $[\Omega]_{m,n} = \mathbb{E}_{\boldsymbol{x} \sim p_X^A} \left[ \left( \frac{\partial \|\phi(\boldsymbol{x};\boldsymbol{\theta})\|^2}{\partial [\boldsymbol{\theta}]_m} \right) \left( \frac{\partial \|\phi(\boldsymbol{x};\boldsymbol{\theta})\|^2}{\partial [\boldsymbol{\theta}]_n} \right) \right]$, which is the importance parameter used by Aljundi et al. (2018). From a geometric perspective, MAS preserves the norms of the samples from

Task A while learning Task B. In other words; MAS enforces closeness of $\boldsymbol{\theta}$ to $\boldsymbol{\theta}_A^*$ in a Riemmanian pseudo-manifold induced by matrix $\Omega$.

The general idea of using the expected value of a "suitable" distance/divergence between samples, $\mathbb{E}_{\boldsymbol{x} \sim p_X^A} d(\phi(\boldsymbol{x}; \boldsymbol{\theta}), \phi(\boldsymbol{x}; \boldsymbol{\theta}_A^*))$ and leveraging its second-order Taylor expansion to obtain synaptic importance values is crucial here and could lead to various undiscovered algorithms based on new distances/divergences. We refer to these approaches as sample-based regularization methods.

### 2.3 Preserving Distribution of an Arbitrary Layer

We approach the problem of overcoming catastrophic forgetting from the angle of preserving Task A's distribution at an arbitrary layer of the neural network, when learning Task B. Let $\boldsymbol{z}_i^A = \phi(\boldsymbol{x}_i^A; \boldsymbol{\theta}) \in \mathbb{R}^K$ be the output of the network, for a sample from Task A, at the target layer (e.g., output logits in a NN classifier, or reconstructed image of an autoencoder). The distribution of the random variable $\boldsymbol{z}^A$ in the target layer follows from the Random Variable Transform (RVT) theorem (Gillespie, 1983):

$$p_Z^A(\boldsymbol{z}|\boldsymbol{\theta}) = \int_{\mathbb{X}} p_X^A(\boldsymbol{x})\delta(\boldsymbol{z} - \phi(\boldsymbol{x}; \boldsymbol{\theta}))dx \tag{8}$$

Then we propose the following general regularization to overcome forgetting when learning task B:

$$\arg\min_{\boldsymbol{\theta}} \tilde{\mathcal{L}}^B(\boldsymbol{\theta}) = \arg\min_{\boldsymbol{\theta}} \mathcal{L}^B(\boldsymbol{\theta}) + \lambda d(p_Z^A(\cdot|\boldsymbol{\theta}), p_Z^A(\cdot|\boldsymbol{\theta}_A^*)) \tag{9}$$

where $d(\cdot, \cdot)$ is a discrepancy measure between the two probability distributions defined in $\mathbb{R}^K$. Note that equation 9 could be performed with any discrepancy measure or distance, e.g., the Wasserstein distance (Villani, 2008; Kolouri et al., 2017), using a memory replay strategy. In what follows, we describe a 'suitable' distance $d(\cdot, \cdot)$ that: 1) respects the underlying geometry of the space, and 2) enables a similar strategy to that of the EWC framework to provide importance parameters.

## 3 Sliced-Cramér Distance for Structural Plasticity

### 3.1 Cramér Distance

The p-Cramér distance (Cramér, 1928; Székely & Rizzo, 2013) between two one-dimensional probability density functions $p_0$ and $p_1$ is defined as the $\ell_p$-norm between their cumulative distribution functions. Note that here we avoid any measure theoretic notations for simplicity. Let $q_i(t) = \int_{-\infty}^t p_i(\tau)d\tau$ denote the cumulative distribution function for $p_i$, then the p-Cramér distance is defined as:

$$C_p(p_0, p_1) = \left( \int_{\mathbb{R}} |q_0(t) - q_1(t)|^p dt \right)^{\frac{1}{p}} \tag{10}$$

for $p \geq 1$. Similar to the Wasserstein distance and unlike the KL-divergence and its symmetric form Jensen-Shannon distance (i.e., the square root of the Jensen-Shannon divergence), the Cramér distance respects the underlying geometry of the space. Moreover, the Cramér distance provides unbiased sample gradients (Bellemare et al., 2017), and for $p = 1$, $C_1$ is equivalent to the 1-Wasserstein distance, $W_1$. In addition, and similar to the Wasserstein metric, the dual of the Cramér distance is of the form of an integral probability metric (IPM) (Dedecker & Merlevède, 2007).

To further demonstrate the favorable characteristics of this distance, consider the following parametric distribution matching in one-dimension, where the target distribution, $p$, is a box distribution defined on $\mathbb{R}$ and $p_\tau(t) = p(t-\tau)$ is the shifted version of $p$ and the goal is to optimize $\tau$ to minimize the distance between $p_\tau$ and $p$ ($\tau^* = 0$). For this simple setting, we calculate the energy landscape (i.e., the distance between $p_\tau$ and $p$) as a function of $\tau$ for the Jensen-Shannon distance, $W_p$, and $C_p$ for $p = 1, 2$. Figure 2 shows the distributions $p$ and $p_\tau$ on the left and the energy landscape as a function of $\tau$ on the right. It can be clearly seen that the Wasserstein and Cramér distances respect the underlying geometry of the problem, while the Jensen-Shannon (JS) distance fails to do so.

To extend the Cramér distance to higher-dimensional distributions, we utilize the idea of distribution slicing used in various recent publications (Kolouri et al., 2016; 2019; 2018). We note that the sliced-Cramér distance (also known as the Cramér-Wold distance) was recently used in Tabor et al. (2018) for generative modeling. We briefly describe the idea in the following section.

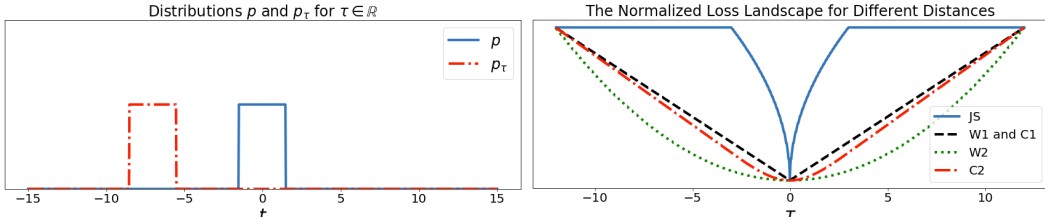

Figure 2: The energy landscape of various distances as a function of the translation parameter. It can be seen that both Wasserstein and Cramér distances respect the underlying geometry of the problem while the Jensen-Shannon distance fails to do so.

## 3.2 SLICED-CRAMÉR DISTANCE

The idea of slicing a higher-dimensional distribution has roots in the Radon transform that is commonly used in computational tomography. The idea is to represent a high-dimensional distribution via the infinite set of its marginal distributions. In short, let $p_0$ and $p_1$ be d-dimensional probability density functions defined on $\mathbb{X} \subset \mathbb{R}^d$, then their Radon transform is defined as:

$$\mathcal{R}p_i(t, \boldsymbol{\xi}) = \int_{\mathbb{X}} p_i(\boldsymbol{x}) \delta(t - \boldsymbol{x} \cdot \boldsymbol{\xi}) dx \tag{11}$$

for $\forall t \in \mathbb{R}$ and $\forall \boldsymbol{\xi} \in \mathbb{S}^{d-1}$ where $\mathbb{S}^{d-1}$ denotes the d-dimensional unit sphere. Note that $\mathcal{R}p_i(\cdot, \boldsymbol{\xi})$ is a so called slice of $p_i$, which is a one-dimensional marginal distribution of $p_i$. Let $\mathcal{R}q_i(\cdot, \boldsymbol{\xi})$ be the corresponding cumulative distribution function of $\mathcal{R}p_i(\cdot, \boldsymbol{\xi})$:

$$\mathcal{R}q_i(t, \boldsymbol{\xi}) = \int_{-\infty}^{t} \mathcal{R}p_i(\tau, \boldsymbol{\xi}) d\tau \tag{12}$$

then the sliced-Cramér distance between $p_0$ and $p_1$ is the expected value of the Cramér distance between their one-dimensional slices, i.e., $\mathcal{R}p_i(\cdot, \boldsymbol{\xi})$ when $\boldsymbol{\xi} \sim \mathcal{U}_{\mathbb{S}^{d-1}}$ for $\mathcal{U}_{\mathbb{S}^{d-1}}$ being the uniform distribution on the d-dimensional unit sphere. In other words, the sliced Cramér distance is defined as:

$$\begin{aligned} SC_p(p_0, p_1) &= \left( \int_{\mathbb{S}^{d-1}} C_p^p(\mathcal{R}p_0(\cdot, \boldsymbol{\xi}), \mathcal{R}p_1(\cdot, \boldsymbol{\xi})) d\boldsymbol{\xi} \right)^{\frac{1}{p}} \\ &= \left( \int_{\mathbb{S}^{d-1}} \int_{\mathbb{R}} |\mathcal{R}q_0(t, \boldsymbol{\xi}) - \mathcal{R}q_1(t, \boldsymbol{\xi})|^p dt d\boldsymbol{\xi} \right)^{\frac{1}{p}} \end{aligned} \tag{13}$$

Now we are ready to propose our method for overcoming representation forgetting.

## 3.3 OVERCOMING REPRESENTATION FORGETTING

The critical point here is that the sample-based regularizers could over-estimate the importance of synapses, leading to intransigence faster. More importantly, a substantial expected change in the network's output (over individual samples) would not necessarily mean catastrophic forgetting. Changes in the network's output within a mode (e.g., in supervised classification within the distribution of a particular class) are harmless, so long as the representation of the data is not significantly changing. Our proposed regularizer tolerates such changes and therefore has the potential for better utilization of the network's learning capacity. We propose the following regularization for overcoming catastrophic forgetting:

$$\arg\min_{\boldsymbol{\theta}} \tilde{\mathcal{L}}^B(\boldsymbol{\theta}) = \arg\min_{\boldsymbol{\theta}} \mathcal{L}^B(\boldsymbol{\theta}) + \lambda SC_2^2(p_Z^A(\cdot|\boldsymbol{\theta}_A^*), p_Z^A(\cdot|\boldsymbol{\theta})) \tag{14}$$

equation 14 requires memory replay from the old task(s) while learning the new one. To transition from memory replay to selective synaptic plasticity, we derive the second-order Taylor expansion of the regularizer $SC_2^2$ around the optimal parameters for the previous tasks, $\theta_A^*$. Assuming that $\boldsymbol{\theta} = \boldsymbol{\theta}_A^* + \delta\boldsymbol{\theta}$ where $\delta\boldsymbol{\theta} \to 0$, it is straightforward to show that:

$$SC_2^2(p_Z^A(\cdot|\boldsymbol{\theta}_A^*), p_Z^A(\cdot|\boldsymbol{\theta})) \approx (\delta\boldsymbol{\theta})^T \Gamma_{\boldsymbol{\theta}_A^*}(\delta\boldsymbol{\theta}) = \|\delta\boldsymbol{\theta}\|_{\Gamma_{\boldsymbol{\theta}_A^*}} \tag{15}$$

where $\Gamma_{\boldsymbol{\theta}_A^*}$ is defined as:

$$\Gamma_{\boldsymbol{\theta}_A^*} := \int_{\mathbb{S}^{d-1}} \int_{\mathbb{R}} \left( \frac{d\, \mathcal{R}q_Z^A(t, \boldsymbol{\xi}|\boldsymbol{\theta}_A^*)}{d\boldsymbol{\theta}} \right) \left( \frac{d\, \mathcal{R}q_Z^A(t, \boldsymbol{\xi}|\boldsymbol{\theta}_A^*)}{d\boldsymbol{\theta}} \right)^T dt d\boldsymbol{\xi} \qquad (16)$$

See supplementary materials for the detailed derivations. Note that similar to $F_{\boldsymbol{\theta}_A^*}$, $\Gamma_{\boldsymbol{\theta}_A^*}$ is also positive-semi-definite (PSD) and therefore our Sliced-Cramér regularizer enforces closeness of $\boldsymbol{\theta}$ and $\boldsymbol{\theta}^*$ in a Riemannian pseudo-manifold induced by the PSD matrix $\Gamma_{\boldsymbol{\theta}}$.

While the definition of $\Gamma$ in equation 16, regardless of its similarity to equation 3, may seem intimidating, it leads to a straightforward empirical algorithm, which we discuss in the next section. Before that, we point out that similar to the FIM, calculating $\Gamma$ is also practically infeasible, and we approximate $\Gamma$ with a diagonal matrix that simplifies the regularization into:

$$\arg\min_{\boldsymbol{\theta}} \tilde{\mathcal{L}}^B(\boldsymbol{\theta}) \quad = \quad \arg\min_{\boldsymbol{\theta}} \mathcal{L}^B(\boldsymbol{\theta}) + \lambda \sum_{m=1}^{M} [\Gamma]_{m,m} [\delta\boldsymbol{\theta}]_m^2 \qquad (17)$$

Finally, we emphasize that while equation 17 is similar to equation 5, it enforces a very different constraint on the neural network. equation 5 enforces conditional class likelihoods to be preserved, which is sample-based; however, our proposed formulation in equation 17 preserves the distribution of network's outputs at a particular layer for old tasks, i.e., it preserves the distribution of the previously learned representations.

## 4    PROPOSED ALGORITHM

Here we derive the algorithmic steps required to calculate $\Gamma$ empirically, as shown in Algorithm 1. The empirical distribution at the network's output can be written as, $p_Z^A(\boldsymbol{z}|\boldsymbol{\theta}) \approx \frac{1}{N} \sum_{n=1}^{N} \delta(\boldsymbol{z} - \phi(\boldsymbol{x}_n^A; \boldsymbol{\theta}))$, and the slices of this empirical distribution are defined as:

$$\mathcal{R}p_Z^A(t, \boldsymbol{\xi}|\boldsymbol{\theta}) \approx \frac{1}{N} \sum_{n=1}^{N} \delta(t - \boldsymbol{\xi} \cdot \phi(\boldsymbol{x}_n^A; \boldsymbol{\theta}))$$

Let $u(\cdot)$ denote the step function, which is the cumulative distribution of the Dirac delta function. Then the cumulative distribution of $\mathcal{R}p_Z^A(t, \boldsymbol{\xi}|\boldsymbol{\theta})$ can be written as, $\mathcal{R}q_Z^A(t, \boldsymbol{\xi}|\boldsymbol{\theta}) \approx \frac{1}{N} \sum_{n=1}^{N} u(t - \boldsymbol{\xi} \cdot \phi(\boldsymbol{x}_n^A; \boldsymbol{\theta}))$. Therefore we have:

$$\frac{d\, \mathcal{R}q_Z^A(t, \boldsymbol{\xi}|\boldsymbol{\theta})}{d\boldsymbol{\theta}} \approx \frac{1}{N} \sum_{n=1}^{N} \left( \frac{-d\, \boldsymbol{\xi} \cdot \phi(\boldsymbol{x}_n^A; \boldsymbol{\theta})}{d\boldsymbol{\theta}} \right) \delta(t - \boldsymbol{\xi} \cdot \phi(\boldsymbol{x}_n^A; \boldsymbol{\theta})) \qquad (18)$$

substituting equation 18 into equation 16 and using a Monte-Carlo approximation of the integration of $\mathbb{S}^{d-1}$, with $L$ samples, leads to:

$$\Gamma = \frac{1}{L} \sum_{l=1}^{L} \left( \frac{d\, \boldsymbol{\xi}_l \cdot \bar{\boldsymbol{z}}}{d\boldsymbol{\theta}} \right) \left( \frac{d\, \boldsymbol{\xi}_l \cdot \bar{\boldsymbol{z}}}{d\boldsymbol{\theta}} \right)^T \qquad (19)$$

where $\boldsymbol{\xi}_l$s are randomly drawn from $\mathbb{S}^{K-1}$, and $\bar{\boldsymbol{z}} = \frac{1}{N} \sum_{n=1}^{N} \phi(\boldsymbol{x}_n^A; \boldsymbol{\theta}_A^*)$. These derivations give birth to our proposed algorithm shown in Algorithm 1. Given that calculation of matrix $\Gamma$ is not practically feasible (due to the large number of parameters of a deep neural network), we follow the work of (Kirkpatrick et al., 2017) and approximate $\Gamma$ to be a diagonal matrix, which simplifies to:

$$[\Gamma_{\boldsymbol{\theta}_A^*}]_{i,i} = \frac{1}{L} \sum_{l=1}^{L} \left( \frac{1}{N} \sum_{n=1}^{N} \frac{d\, \boldsymbol{\xi}_l \cdot \phi(\boldsymbol{x}_n^A; \boldsymbol{\theta}_A^*)}{d\boldsymbol{\theta}_i} \right)^2 = \frac{1}{L} \sum_{l=1}^{L} \left( \frac{d\, \boldsymbol{\xi}_l \cdot \bar{\phi}_A^*}{d\boldsymbol{\theta}_i} \right)^2$$

where $\bar{\phi}_A^* = \frac{1}{N} \sum_{n=1}^{N} \phi(\boldsymbol{x}_n^A; \boldsymbol{\theta}_A^*)$.

---

**Algorithm 1:** Sliced Cramer Preservation (SCP)

**Input:** Data, $\mathcal{X}^A = \{\boldsymbol{x}_n^A \sim p_X^A\}_{n=1}^N$, and the optimized neural network, $\phi(\cdot; \boldsymbol{\theta}_A^*)$, for Task $A$.
**Parameters:** Number of random projections, $L$.
**Output:** Synaptic importance matrix $\Gamma$

1 Calculate the mean response of the network at the targeted layer: $\bar{\phi}_A^* = \frac{1}{N}\sum_{n=1}^N \phi(\boldsymbol{x}_n^A; \theta_A^*)$.
2 Initialize the synaptic importance matrix, $[\Gamma]_{i,j} = 0$.
3 **for** $l \leftarrow 1$ **to** $L$ **do**
4 $\quad$ Sample $\boldsymbol{\xi}_l$ from $\mathbb{S}^{K-1}$
5 $\quad$ Slice the mean response: $\rho = \boldsymbol{\xi}_l \cdot \bar{\phi}_A^*$
6 $\quad$ Calculate $\nabla_{\boldsymbol{\theta}}\rho$ using auto-differentiation
7 $\quad$ Update $\Gamma$: $\Gamma \mathrel{+}= \frac{1}{L}(\nabla_{\boldsymbol{\theta}}\rho)(\nabla_{\boldsymbol{\theta}}\rho)^T$

---

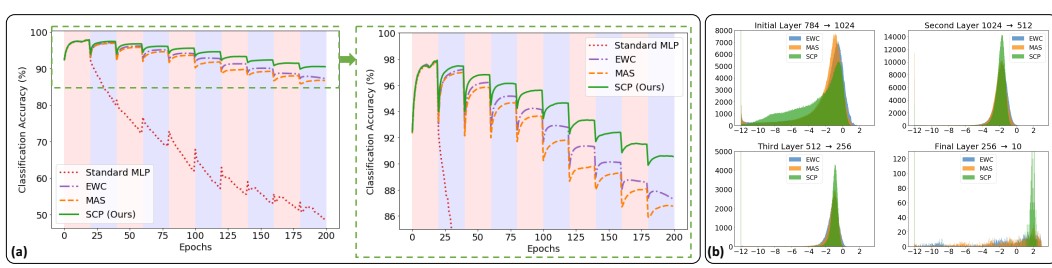

Figure 3: Comparison among online-EWC, MAS, and SCP on learning ten permuted MNIST tasks (a), and the histogram of $Log(\lambda \times \cdot)$ of the synaptic importances (i.e., the diagonal values of $F$, $\Omega$, and $\Gamma$, for EWC, MAS, and SCP, respectively) for each method (b).

### 4.1 ONLINE EXTENSION OF THE ALGORITHM

To extend the framework into sequential learning of multiple tasks and to avoid memorizing task-specific $\Gamma$s (or their diagonals), we follow the EWC++ framework proposed by Chaudhry et al. (2018), which is, in essence, identical to the online-EWC proposed by Schwarz et al. (2018). The EWC++ (and online-EWC) methods ameliorate the need for predicting task identities and calculating task-specific FIMs and keep the memory requirement of the method constant (EWC requires linear growth of memory as a function of number of tasks). In EWC++, given $F_\theta^{(t-1)}$ at task $(t-1)$, the accumulated FIM after learning task $t$ is calculated as $F_{\boldsymbol{\theta}}^{(t)} = \alpha F_{\boldsymbol{\theta}_t^*} + (1-\alpha)F_{\boldsymbol{\theta}}^{(t-1)}$, where $F_{\boldsymbol{\theta}_t^*}$ is the task specific FIM for task $t$, and $\alpha \in [0,1)$ is a hyperparameter that indicates the importance of preserving the most recent task over the older ones. Similarly, we use,

$$\Gamma_{\boldsymbol{\theta}}^{(t)} = \alpha\Gamma_{\boldsymbol{\theta}_t^*} + (1-\alpha)\Gamma_{\boldsymbol{\theta}}^{(t-1)} \tag{20}$$

to obtain the sliced-Cramér regularizer for task $(t+1)$.

## 5 NUMERICAL EXPERIMENTS

### 5.1 PERMUTED MNIST

We first test our proposed algorithm on the benchmark permuted MNIST task and compare the performance with online-EWC and our implementation of the online-MAS algorithm. For this experiment, we used a single head model that learns ten tasks, where each task contains a permuted version of the original MNIST dataset. We note that the reported results are a function of the architecture of the underlying network. Meaning that, while all three methods perform well on this task for larger networks, the true competitiveness of the proposed method emerges for smaller networks where "over-estimation" of the synaptic importances significantly hinders learning of the subsequent tasks and leads to intransigence. For this experiment, we used a fully-connected network (i.e.,

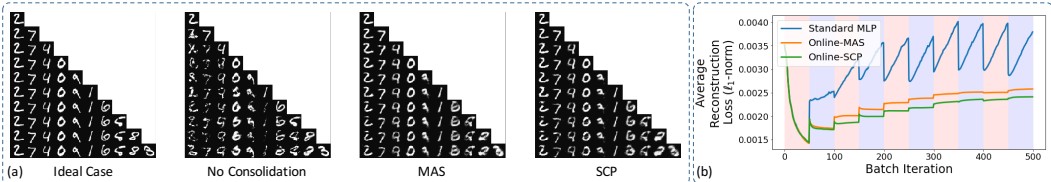

Figure 4: Qualitative and quantitative comparison between MAS and SCP on sequential learning of auto-encoders. Columns in Panel (a) show the reconstruction of data after learning consequent tasks in a random permutation of MNIST sequence. Panel (b) shows the average $\ell_1$-reconstruction loss for each method over all tasks and over 10 runs.

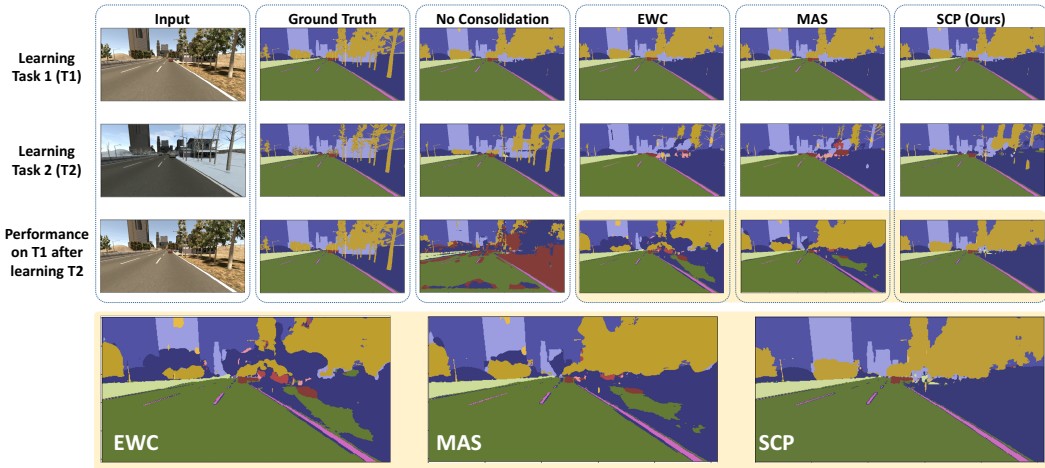

Figure 5: A qualitative comparison of the EWC, MAS, and SCP algorithms on semantic segmentation of the SYNTHIA Dataset (Ros et al., 2016). Task 1 (T1) is semantic segmentation in summer, and Task 2 (T2) is semantic segmentation in winter. The first row shows performance after learning T1 on input from T1. Second row, shows performance after learning T2 on input from T2. The third row shows performance on input from T1 after learning T2. The last row, magnifies the last three images in the third row for the ease of comparison.

a multi-layer perceptron) with the following architecture, $784 \rightarrow 1024 \rightarrow 512 \rightarrow 256 \rightarrow 10$ neurons, and for all optimizations we used the ADAM optimizer with learning rate, $lr = 1e - 4$. For our proposed method, SCP, we used $L = 100$ slices. We repeated each experiment 10 times (with different permutations), and reported the average accuracy over all tasks in Figure 3 (a). We can see that SCP is capable of utilizing the capacity of the network better, which points to the fact that regularizing the distribution as opposed to the samples provides a less restricted regularization for the network and still enables the network to freely move individual samples so long as the overall latent distribution of the data is consistent.

The regularization coefficients for each algorithm was cross-validated on the following grid, $\lambda \in \{1e + i \mid i \in [-3, -2, ..., 9]\}$ and the optimal value was used to report the results in Figure 3 (a). Moreover, in Figure 3 (b) we show the histogram of the logarithm of the product of the regularization coefficients with importance parameters for each method (i.e., $\lambda$ times the diagonal values of $F_\theta$, $\Omega$, and $\Gamma$ for EWC, MAS, and SCP, respectively). One can see that the optimal values of the regularization coefficients provide, more or less, scale-consistent synaptic importances for all methods and distinguishing factor between the methods is on the difference between these distributions. Another interesting observation is that the distribution of the synaptic importances are more similar for sample-based methods (i.e., EWC and MAS) compared to the proposed distribution-based method.

## 5.2 SEQUENTIAL LEARNING OF AUTO-ENCODERS

Next, we consider an experiment consisting of unsupervised/self-supervised sequential learning. To that end, we learn an auto-encoder on single digits of the MNIST dataset sequentially. The model is chosen to be a fully connected auto-encoder, with the following encoder $728 \rightarrow 1024 \rightarrow 1024 \rightarrow 1024 \rightarrow 256$, a mirrored decoder $256 \rightarrow 1024 \rightarrow 1024 \rightarrow 1024 \rightarrow 784$, and Rectified Linear Unit (ReLU) activations. For the loss function, we used cross-entropy plus the $\ell_1$-norm of the reconstruction error. Similar to the previous experiment, we used the ADAM optimizer (Kingma & Ba, 2014) for training the network with $lr = 1e-4$. We perform 50 epochs of learning on each digit, before switching to the next digit. For consolidation, we used $L = 100$ slices for SCP. Finally, we permute the order of the digits and run our experiments 10 times, and compare 'No Consolidation,' with MAS, and SCP. Due to the unsupervised nature of the experiment, the EWC framework does not apply here.

Figure 4 demonstrate the results of this experiment. Panel (a) shows the reconstruction of a sample digit from one of our runs with the input digit sequence of $[2, 7, 4, 0, 9, 1, 6, 5, 8, 3]$. It can be seen that both MAS and SCP can successfully retain the learned knowledge while acquiring new knowledge, while without consolidation the auto-encoder suffers from catastrophic forgetting. Panel (b) shows the average $\ell_1$-norm of the reconstruction error for all seen tasks over the 10 runs. We can see that SCP and MAS are qualitatively on par, and SCP provides a modest yet statistically significant improvement over MAS for this task.

## 5.3 SEMANTIC SEGMENTATION OF SYNTHIA DATASET

Lastly, we go beyond the benchmark yet less practical MNIST dataset and address catastrophic forgetting in a more interesting/critical application of autonomous vehicles. We specifically consider the problem of learning semantic segmentation of road scenes in a sequential manner, where the input distribution of the data changes over time. Semantic segmentation is the task of assigning a class label to every pixel of an input image. To that end, we use two sequences of the SYNTHIA dataset (Ros et al., 2016), namely 'SYNTHIA-SEQS-01-SUMMER' and 'SYNTHIA-SEQS-01-WINTER' as Task 1 and Task 2, respectively. There are 13 classes in the dataset namely: Miscellaneous, Sky, Building, Road, Sidewalk, Fence, Vegetation, Pole, Car, Sign, Pedestrian, Cyclist, and Lane Marking. We keep the last 100 frames of each sequence as the testing-set and train a deep convolutional U-Net architecture (Ronneberger et al., 2015) on the tasks mentioned above. For the loss function, we used $(1 - Dice)$ (Zou et al., 2004), and each task was learned over 100 epochs. For the optimizer, we used the ADAM optimizer (Kingma & Ba, 2014) with learning rate, $lr = 1e - 4$. For consolidation, we used $L = 100$ slices for SCP.

We learn the tasks sequentially (summer first and then winter), and a qualitative comparison of online-EWC, MAS, and SCP on a sample test frame is in Figure 5. The first row shows the performance on T1 after learning T1 (base model), the second row shows performance on T2 after learning on T1 and then T2 (indicator of intransigence), and the third row shows performance on T1 after learning on T1 and then T2 (indicator of the catastrophic forgetting). We can see that catastrophic forgetting happens when no synaptic consolidation is leveraged. Moreover, compared to online-EWC and MAS, SCP suffers less from intransigence as fewer artifacts are present in the second row. Lastly, SCP overcomes catastrophic-forgetting more successfully compared to EWC and MAS as it is apparent from the lack of artifacts in the last row. Finally, we provide a quantitative comparison between the methods in Figure 6, where we report the Dice score (Zou et al., 2004), averaged over ten runs, for each task and each method during the sequential training. As

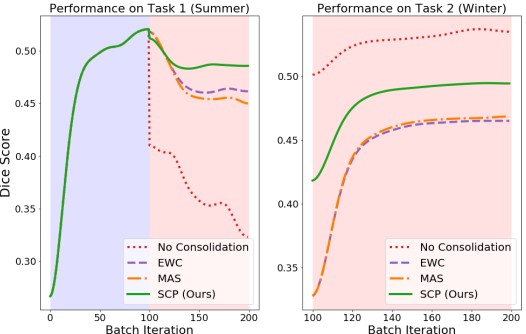

Figure 6: Testing Dice score (Zou et al., 2004) of online-EWC, MAS, and SCP on sequential learning for semantic segmentation of the summer images (Task 1) and winter images (Task 2) from SYNTHIA dataset (Ros et al., 2016). The blue and red shadings on the plots indicate the durations in which the models were trained on summer and winter data, respectively

can be seen, SCP significantly outperforms online-EWC and MAS on this task both in overcoming catastrophic forgetting (left plot) and overcoming intransigence (right plot).

# 6    CONCLUSION

We introduced a new generic approach towards selective synaptic plasticity for preserving the distribution of the network's output, at an arbitrary layer, for previously learned tasks. We started from a memory-replay-based regularization that penalized the change in the distribution of the network's output and showed that a second-order Taylor expansion of such regularization would lead to a selective synaptic plasticity approach that does not require memory of samples from previously seen tasks. Furthermore, we proposed the sliced-Cramér distance as a suitable metric for preserving these distributions, which leads to a straightforward algorithm for selective plasticity. Also, using a similar approach, we reverse-engineered the Memory Aware Synapses (MAS) framework and provided a geometrically meaningful regularization that leads to this algorithm. We then compared the online-EWC, MAS, and SCP methods on a variety of learning tasks, including supervised and unsupervised/self-supervised learning, and consistently showed competitive performance.

# 7    ACKNOWLEDGMENTS

This material is based upon work supported by the United States Air Force and DARPA under Contract No. FA8750-18-C-0103. Any opinions, findings and conclusions or recommendations expressed in this material are those of the author(s) and do not necessarily reflect the views of the United States Air Force and DARPA.

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

## 8 SUPPLEMENTARY MATERIALS

### 8.1 TAYLOR EXPANSION OF THE KL-DIVERGENCE

For the sake of completion, here we derive the second-order Taylor expansion of $D_{\mathrm{KL}}$,

$$D_{\mathrm{KL}}(p_{\boldsymbol{\theta}_0}||p_{\boldsymbol{\theta}}) = \int_X p_{\boldsymbol{\theta}_0}(x) log\left(\frac{p_{\boldsymbol{\theta}_0}(x)}{p_{\boldsymbol{\theta}}(x)}\right) dx$$

around $\boldsymbol{\theta}_0$ where we can write $\boldsymbol{\theta} = \boldsymbol{\theta}_0 + \delta\boldsymbol{\theta}$. The second-order Taylor expansion is:

$$D_{\mathrm{KL}}(p_{\boldsymbol{\theta}_0}||p_{\boldsymbol{\theta}}) \approx D_{\mathrm{KL}}(p_{\boldsymbol{\theta}_0}||p_{\boldsymbol{\theta}_0}) + \delta\boldsymbol{\theta}^T\left(\frac{dD_{\mathrm{KL}}(p_{\boldsymbol{\theta}_0}||p_{\boldsymbol{\theta}})}{d\boldsymbol{\theta}}|_{\boldsymbol{\theta}_0}\right) + \frac{1}{2}\delta\boldsymbol{\theta}^T\left(\frac{d^2D_{\mathrm{KL}}(p_{\boldsymbol{\theta}_0}||p_{\boldsymbol{\theta}})}{d\boldsymbol{\theta}^2}|_{\boldsymbol{\theta}_0}\right)\delta\boldsymbol{\theta}$$

1. Where for the first-order term we have:

$$\frac{dD_{\mathrm{KL}}(p_{\boldsymbol{\theta}_0}||p_{\boldsymbol{\theta}})}{d\boldsymbol{\theta}} = -\int_X p_{\boldsymbol{\theta}_0}(x)\frac{d\,log(p_{\boldsymbol{\theta}}(x))}{d\boldsymbol{\theta}}dx = -\int_X \frac{p_{\boldsymbol{\theta}_0}(x)}{p_{\boldsymbol{\theta}}(x)}\frac{dp_{\boldsymbol{\theta}}(x)}{d\boldsymbol{\theta}}dx$$

Therefore at $\boldsymbol{\theta} = \boldsymbol{\theta}_0$ we have:

$$\frac{dD_{\mathrm{KL}}(p_{\boldsymbol{\theta}_0}||p_{\boldsymbol{\theta}})}{d\boldsymbol{\theta}}|_{\boldsymbol{\theta}_0} = -\int_X \frac{dp_{\boldsymbol{\theta}}(x)}{d\boldsymbol{\theta}}dx = \frac{d}{d\boldsymbol{\theta}}\left(\int_X p_{\boldsymbol{\theta}_0}(x)dx\right) = 0$$

2. and for the second-order term:

$$\begin{aligned}
\frac{d^2D_{\mathrm{KL}}(p_{\boldsymbol{\theta}_0}||p_{\boldsymbol{\theta}})}{d\boldsymbol{\theta}^2} &= \frac{d}{d\boldsymbol{\theta}}\left(-\int_X \frac{p_{\boldsymbol{\theta}_0}(x)}{p_{\boldsymbol{\theta}}(x)}\frac{dp_{\boldsymbol{\theta}}(x)}{d\boldsymbol{\theta}}dx\right) \\
&= \int_X \frac{p_{\boldsymbol{\theta}_0}(x)}{p_{\boldsymbol{\theta}}^2(x)}\left(\frac{dp_{\boldsymbol{\theta}}(x)}{d\boldsymbol{\theta}}\right)\left(\frac{dp_{\boldsymbol{\theta}}(x)}{d\boldsymbol{\theta}}\right)^T dx - \\
&\quad \int_X \frac{p_{\boldsymbol{\theta}_0}(x)}{p_{\boldsymbol{\theta}}(x)}\frac{d^2\,p_{\boldsymbol{\theta}}(x)}{d\boldsymbol{\theta}^2}dx
\end{aligned}$$

Therefore at $\boldsymbol{\theta} = \boldsymbol{\theta}_0$ we have:

$$\begin{aligned}
\frac{d^2D_{\mathrm{KL}}(p_{\boldsymbol{\theta}_0}||p_{\boldsymbol{\theta}})}{d\boldsymbol{\theta}^2}|_{\boldsymbol{\theta}_0} &= \int_X p_{\boldsymbol{\theta}_0}(x)\left(\frac{1}{p_{\boldsymbol{\theta}_0}(x)}\frac{dp_{\boldsymbol{\theta}_0}(x)}{d\boldsymbol{\theta}}\right)\left(\frac{1}{p_{\boldsymbol{\theta}_0}(x)}\frac{dp_{\boldsymbol{\theta}_0}(x)}{d\boldsymbol{\theta}}\right)^T dx - \\
&\quad \int_X \frac{d^2\,p_{\boldsymbol{\theta}_0}(x)}{d\boldsymbol{\theta}^2}dx \\
&= \int_X p_{\boldsymbol{\theta}_0}(x)\left(\frac{d\,log(p_{\boldsymbol{\theta}_0}(x))}{d\boldsymbol{\theta}}\right)\left(\frac{d\,log(p_{\boldsymbol{\theta}_0}(x))}{d\boldsymbol{\theta}}\right)^T dx \\
&= \mathbb{E}_{x\sim p_{\boldsymbol{\theta}_0}}\left[\left(\frac{d\,log(p_{\boldsymbol{\theta}_0}(x))}{d\boldsymbol{\theta}}\right)\left(\frac{d\,log(p_{\boldsymbol{\theta}_0}(x))}{d\boldsymbol{\theta}}\right)^T\right] = F
\end{aligned}$$

which is the Fisher Information Matrix (FIM).

Finally, putting everything together we have:

$$D_{\mathrm{KL}}(p_{\boldsymbol{\theta}_0}||p_{\boldsymbol{\theta}}) \approx \frac{1}{2}\delta\boldsymbol{\theta}^T F\delta\boldsymbol{\theta}$$

which concludes the derivation.

### 8.2 TAYLOR EXPANSION OF THE MAS REGULARIZER

Here we drive the second-order Taylor expansion of the MAS regularizer in equation 6,

$$\begin{aligned}
MAS_{reg} &= \mathbb{E}_{\boldsymbol{x}\sim p_X}\left[\frac{1}{2}(\|\phi(\boldsymbol{x};\boldsymbol{\theta})\|^2 - \|\phi(\boldsymbol{x};\boldsymbol{\theta}_0)\|^2)^2\right] \\
&= \frac{1}{2}\int_X p_X(\boldsymbol{x})(\|\phi(\boldsymbol{x};\boldsymbol{\theta})\|^2 - \|\phi(\boldsymbol{x};\boldsymbol{\theta}_0)\|^2)^2 d\boldsymbol{x}
\end{aligned}$$

around $\boldsymbol{\theta}_0$. The second-order Taylor expansion is:

$$MAS_{reg} \approx \delta\boldsymbol{\theta}^T\left(\frac{d\,MAS_{reg}}{d\boldsymbol{\theta}}|_{\boldsymbol{\theta}_0}\right) + \frac{1}{2}\delta\boldsymbol{\theta}^T\left(\frac{d^2MAS_{reg}}{d\boldsymbol{\theta}^2}|_{\boldsymbol{\theta}_0}\right)\delta\boldsymbol{\theta}$$

1. Where for the first-order term we have:

$$\left[\frac{dMAS_{reg}}{d\boldsymbol{\theta}}\right]_m = \int_X p_X(\boldsymbol{x})\frac{d\,\|\phi(\boldsymbol{x};\boldsymbol{\theta})\|^2}{d\,[\boldsymbol{\theta}]_m}(\|\phi(\boldsymbol{x};\boldsymbol{\theta})\|^2 - \|\phi(\boldsymbol{x};\boldsymbol{\theta}_0)\|^2)$$

which for $\boldsymbol{\theta} = \boldsymbol{\theta}_0$ is zero.

2. For the second-order term we have:

$$\left[\frac{d^2MAS_{reg}}{d\boldsymbol{\theta}^2}\right]_{m,n} = \int_X p_X(\boldsymbol{x})\frac{d\,\|\phi(\boldsymbol{x};\boldsymbol{\theta})\|^2}{d\,[\boldsymbol{\theta}]_m}\frac{d\,\|\phi(\boldsymbol{x};\boldsymbol{\theta})\|^2}{d\,[\boldsymbol{\theta}]_n}d\boldsymbol{x} +$$
$$\int_X p_X(\boldsymbol{x})\left[\frac{d^2\,\|\phi(\boldsymbol{x};\boldsymbol{\theta})\|^2}{d\,\boldsymbol{\theta}^2}\right]_{m,n}(\|\phi(\boldsymbol{x};\boldsymbol{\theta})\|^2 - \|\phi(\boldsymbol{x};\boldsymbol{\theta}_0)\|^2)d\boldsymbol{x}$$

where evaluated at $\boldsymbol{\theta} = \boldsymbol{\theta}_0$ the second term on the right-hand-side vanishes.

Finally, putting everything together we have:

$$MAS_{reg} \approx \delta\boldsymbol{\theta}^T \underbrace{\mathbb{E}_{\boldsymbol{x}\sim p_X}\left[\left(\frac{d\,\|\phi(\boldsymbol{x};\boldsymbol{\theta})\|^2}{d\,\boldsymbol{\theta}}\right)\left(\frac{d\,\|\phi(\boldsymbol{x};\boldsymbol{\theta})\|^2}{d\,\boldsymbol{\theta}}\right)^T\right]}_{\Omega}\delta\boldsymbol{\theta}$$

which is the term reported in the paper.

## 8.3 TAYLOR EXPANSION OF THE SLICED-CRAMÉR DISTANCE

Here we derive the second-order Taylor expansion of the squared 2-Sliced-Cramér distance,

$$SC_2^2(p_{\boldsymbol{\theta}_0}, p_{\boldsymbol{\theta}}) = \int_{\mathbb{S}^{d-1}}\int_{\mathbb{R}}|\mathcal{R}q_{\boldsymbol{\theta}_0}(t,\boldsymbol{\xi}) - \mathcal{R}q_{\boldsymbol{\theta}}(t,\boldsymbol{\xi})|^2 dtd\boldsymbol{\xi}$$

around $\boldsymbol{\theta}_0$, as reported in equation 16. The second-order Taylor expansion is:

$$SC_2^2(p_{\boldsymbol{\theta}_0}, p_{\boldsymbol{\theta}}) \approx SC_2^2(p_{\boldsymbol{\theta}_0}, p_{\boldsymbol{\theta}_0}) + \delta\boldsymbol{\theta}^T\left(\frac{d\,SC_2^2(p_{\boldsymbol{\theta}_0}, p_{\boldsymbol{\theta}})}{d\boldsymbol{\theta}}|_{\boldsymbol{\theta}_0}\right) + \frac{1}{2}\delta\boldsymbol{\theta}^T\left(\frac{d^2SC_2^2(p_{\boldsymbol{\theta}_0}, p_{\boldsymbol{\theta}})}{d\boldsymbol{\theta}^2}|_{\boldsymbol{\theta}_0}\right)\delta\boldsymbol{\theta}$$

1. Where for the first-order term we have:

$$\frac{d\,SC_2^2(p_{\boldsymbol{\theta}_0}, p_{\boldsymbol{\theta}})}{d\boldsymbol{\theta}} = 2\int_{\mathbb{S}^{d-1}}\int_{\mathbb{R}}\frac{d\,\mathcal{R}q_{\boldsymbol{\theta}}(t,\boldsymbol{\xi})}{d\boldsymbol{\theta}}(\mathcal{R}q_{\boldsymbol{\theta}}(t,\boldsymbol{\xi}) - \mathcal{R}q_{\boldsymbol{\theta}_0}(t,\boldsymbol{\xi}))dtd\boldsymbol{\xi}$$

which for $\boldsymbol{\theta} = \boldsymbol{\theta}_0$ is equal to zero.

2. For the second-order term we have:

$$\frac{d^2\,SC_2^2(p_{\boldsymbol{\theta}_0}, p_{\boldsymbol{\theta}})}{d\boldsymbol{\theta}^2} = 2\int_{\mathbb{S}^{d-1}}\int_{\mathbb{R}}\frac{d^2\,\mathcal{R}q_{\boldsymbol{\theta}}(t,\boldsymbol{\xi})}{d\boldsymbol{\theta}^2}(\mathcal{R}q_{\boldsymbol{\theta}}(t,\boldsymbol{\xi}) - \mathcal{R}q_{\boldsymbol{\theta}_0}(t,\boldsymbol{\xi}))dtd\boldsymbol{\xi} +$$
$$2\int_{\mathbb{S}^{d-1}}\int_{\mathbb{R}}\left(\frac{d\,\mathcal{R}q_{\boldsymbol{\theta}}(t,\boldsymbol{\xi})}{d\boldsymbol{\theta}}\right)\left(\frac{d\,\mathcal{R}q_{\boldsymbol{\theta}}(t,\boldsymbol{\xi})}{d\boldsymbol{\theta}}\right)^T dtd\boldsymbol{\xi}$$

which evaluated at $\boldsymbol{\theta} = \boldsymbol{\theta}_0$ is:

$$\frac{d^2\,SC_2^2(p_{\boldsymbol{\theta}_0}, p_{\boldsymbol{\theta}})}{d\boldsymbol{\theta}^2}|_{\boldsymbol{\theta}_0} = 2\int_{\mathbb{S}^{d-1}}\int_{\mathbb{R}}\left(\frac{d\,\mathcal{R}q_{\boldsymbol{\theta}_0}(t,\boldsymbol{\xi})}{d\boldsymbol{\theta}}\right)\left(\frac{d\,\mathcal{R}q_{\boldsymbol{\theta}_0}(t,\boldsymbol{\xi})}{d\boldsymbol{\theta}}\right)^T dtd\boldsymbol{\xi}$$
$$= 2\Gamma$$

Finally, putting everything together we have:

$$SC_2^2(p_{\boldsymbol{\theta}_0}, p_{\boldsymbol{\theta}}) \approx \delta\boldsymbol{\theta}^T\Gamma\delta\boldsymbol{\theta}$$

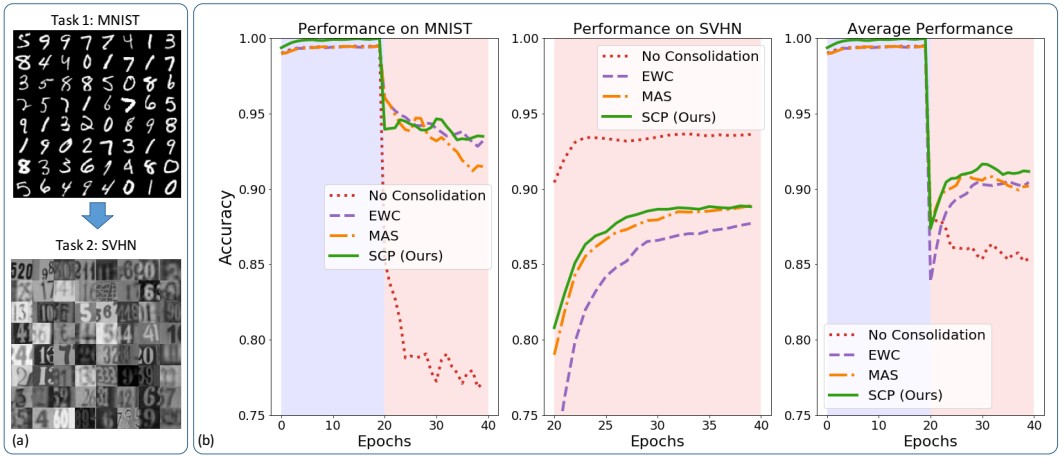

Figure 7: Sample images from MNIST and SVHN dataset (a), and the testing performance of the Online-EWC, MAS, and SCP on sequential learning starting from MNIST (Task 1) and then SVHN (Task 2) (b). The blue and red shadings on the plots indicate the durations in which the models were trained on the MNIST and SVHN datasets, respectively.

```
VGG_like(
  (activation): ReLU(inplace)
  (features): Sequential(
    (0): Conv2d(1, 64, kernel_size=(3, 3), stride=(1, 1), padding=(2, 2))
    (1): BatchNorm2d(64, eps=1e-05, momentum=0.1, affine=True, track_running_stats=True)
    (2): ReLU(inplace)
    (3): Conv2d(64, 64, kernel_size=(3, 3), stride=(1, 1), padding=(2, 2))
    (4): BatchNorm2d(64, eps=1e-05, momentum=0.1, affine=True, track_running_stats=True)
    (5): ReLU(inplace)
    (6): MaxPool2d(kernel_size=2, stride=2, padding=0, dilation=1, ceil_mode=False)
    (7): Conv2d(64, 128, kernel_size=(3, 3), stride=(1, 1), padding=(2, 2))
    (8): BatchNorm2d(128, eps=1e-05, momentum=0.1, affine=True, track_running_stats=True)
    (9): ReLU(inplace)
    (10): Conv2d(128, 128, kernel_size=(3, 3), stride=(1, 1), padding=(2, 2))
    (11): BatchNorm2d(128, eps=1e-05, momentum=0.1, affine=True, track_running_stats=True)
    (12): ReLU(inplace)
    (13): MaxPool2d(kernel_size=2, stride=2, padding=0, dilation=1, ceil_mode=False)
    (14): Conv2d(128, 256, kernel_size=(3, 3), stride=(1, 1), padding=(2, 2))
    (15): BatchNorm2d(256, eps=1e-05, momentum=0.1, affine=True, track_running_stats=True)
    (16): ReLU(inplace)
    (17): Conv2d(256, 256, kernel_size=(3, 3), stride=(1, 1), padding=(2, 2))
    (18): BatchNorm2d(256, eps=1e-05, momentum=0.1, affine=True, track_running_stats=True)
    (19): ReLU(inplace)
    (20): MaxPool2d(kernel_size=2, stride=2, padding=0, dilation=1, ceil_mode=False)
  )
  (fc): Sequential(
    (0): Linear(in_features=12544, out_features=2048, bias=True)
    (1): BatchNorm1d(2048, eps=1e-05, momentum=0.1, affine=True, track_running_stats=True)
    (2): ReLU(inplace)
    (3): Dropout(p=0.2)
    (4): Linear(in_features=2048, out_features=1024, bias=True)
    (5): BatchNorm1d(1024, eps=1e-05, momentum=0.1, affine=True, track_running_stats=True)
    (6): ReLU(inplace)
    (7): Dropout(p=0.2)
  )
  (classify): Sequential(
    (0): Linear(in_features=1024, out_features=10, bias=True)
  )
)
```

Figure 8: The model we used in the MNIST-to-SVHN experiment.

## 8.4 MNIST-TO-SVHN EXPERIMENT

Given the space constraint of the conference, we include our results on the MNIST-to-SVHN experiment in the supplementary material. In this experiment, Task 1 is learning the MNIST digits, and Task 2 is learning the SVHN digits. We show sample images from these two datasets in Figure 7a. In this experiment, we used a VGG-like architecture, for which we include the details in 8.

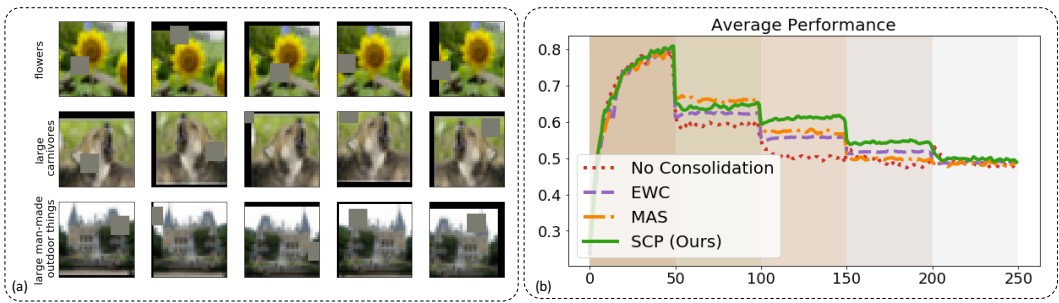

Figure 9: Sample super-classes from the CIFAR100 dataset and visualization of the data augmentation procedure we used in this experiment (a), and the average testing accuracy of the Online-EWC, MAS, and SCP on sequential learning of the tasks (b). We note that we were, unfortunately, only able to run the experiment once, and will update the Figure with the results from multiple runs as soon as they are available.

We performed incremental learning with no consolidation, Online-EWC, MAS, and SCP. The importance parameters were cross-validated on a coarse grid (due to the computational and time restrictions) of $\{1e + i | i \in [2, 3, 4, 5, 6]\}$, where the optimal parameters for the methods where $1e + 3$ for SCP, $1e + 5$ for MAS, and $1e + 4$ for Online-EWC. We chose 20 epochs per task. Figure 7b shows the results of the experiment. Note that the first plot shows the testing performance of the methods on Task1 (i.e., MNIST dataset), where the blue shade indicates the first 20 epochs (training on the MNIST dataset), and the red shade indicates the second 20 epochs (training on the SVHN dataset). As can be seen, all methods can address catastrophic forgetting, while Online-EWC and SCP outperform MAS (However, a one should perform a finer grid search on the importance-parameters for a definitive evaluation). The middle plot shows the testing performance of the model on the SVHN dataset (Task 2). The MAS and SCP methods outperform online-EWC, and SCP performs slightly better than MAS (i.e., overcomes intransigence better). The third plot shows the average testing performance of the methods on both tasks. Similarly, we can see that SCP outperforms MAS and Online-EWC. We repeated the experiments ten times, and the plots show the average performance.

## 8.5 CIFAR100 EXPERIMENTS

The CIFAR100 (Krizhevsky et al., 2009) dataset contains images from 20 super-classes, where each super-class contains five sub-classes. For this dataset, we considered an experiment in which the tasks are supervised classification of the super-classes. In short, we split the data into five sequential tasks, where each task contains a sub-class from all the 20 super-classes. We used a Wide Residual Network (Wide-ResNet) Zagoruyko & Komodakis (2016) network as our model and utilized the data augmentation suggested by (DeVries & Taylor, 2017). Figure 9a shows sample super-classes together with data augmentation. For each method, we used a coarse grid search for the importance parameters $\{1e + i \mid i \in [2, 3, 4, 5, 6]\}$. Unfortunately, and due to the lack of time, we were able to only run the experiment for each method once. We report the average classification results in Figure 9b. For the final version of the paper, we will complete this experiment with 1) a finer grid search over the importance parameters and 2) multiple runs of the experiments and reporting the average.

