# OpenReview forum: "Sliced Cramer Synaptic Consolidation for Preserving Deeply Learned Representations"
_ICLR.cc/2020/Conference — Accept (Spotlight)_

### Official Review · AnonReviewer1 · 2019-10-24
**Official Blind Review #1**

**Rating:** 8

**Review:**

I think the paper is written quite well, and the approach makes a lot of sense. I think the idea of replacing generalizing the KL to sliced Cramer distance is quite interested. The authors put in some effort in explaining why they propose this alternative distance between distributions.

I think overall this is a great paper, very informative. If I need to nitpick, I think the experimental section relies heavily on MNIST (e.g. permuted MNIST, auto-encoding MNIST), which I think is not a hard enough task. It is though widely used in the continual learning community, though maybe it should not anymore. But I think given the reliance of the field on these datasets it makes sense, plus I think the strength of the paper is not in the empirical evaluation but rather in the derivation of the method.

I think while the authors put quite a bit of effort in explaining the difference between KL and Cramer distance, I would have appreciated a even more detailed exposition. I think the difference between these metrics is not well understood by the majority in the community.

**Experience Assessment:**

I have published in this field for several years.

**Review Assessment: Checking Correctness Of Derivations And Theory:**

I did not assess the derivations or theory.

**Review Assessment: Checking Correctness Of Experiments:**

I assessed the sensibility of the experiments.

**Review Assessment: Thoroughness In Paper Reading:**

I read the paper at least twice and used my best judgement in assessing the paper.

---

> ### Author Response · Authors · 2019-11-15
> **Response to Official Blind Review #1**
>
> Thank you very much for a positive evaluation of our work and for setting time aside to carefully read our paper. We appreciate your valuable time spent on serving the community. Below please find our responses.
>
>
> 1) "I think the experimental section relies heavily on MNIST (e.g. permuted MNIST, auto-encoding MNIST), which I think is not a hard enough task. It is though widely used in the continual learning community, though maybe it should not anymore. But I think given the reliance of the field on these datasets it makes sense, plus I think the strength of the paper is not in the empirical evaluation but rather in the derivation of the method."
>
> We agree with the reviewer that our experimental section heavily relied on the MNIST dataset. Reviewer #3 also raised the same point, however, as you mentioned the main strength of the paper is certainly not in our empirical evaluation. To that end,  we added the experiments requested by Reviewer #3 on MNIST-to-SVHN and sequential learning on CIFAR-100. We demonstrate that the results on the new datasets are consistent with the ones reported in the original submission. Please see the extended supplementary material in the revised manuscript.
>
> 2) "I think while the authors put quite a bit of effort in explaining the difference between KL and Cramer distance, I would have appreciated a even more detailed exposition. I think the difference between these metrics is not well understood by the majority in the community."
>
> This is a great point. Although we are restricted by the page limit of the conference, we did our best to revise the manuscript to further clarify the characteristics of these metrics and made additional connections to integral probability metrics (IPMs).
>
> Yet again we thank the reviewer for a candid evaluation of our work.

---

### Official Review · AnonReviewer2 · 2019-10-27
**Official Blind Review #3**

**Rating:** 6

**Review:**

[Summary]
This paper proposes a new method for overcoming catastrophic forgetting in continual learning, based on distribution-based regularization using the sliced Cramer distance, i.e. Sliced Cramer Preservation (SCP). Unlike previous work on catastrophic forgetting, this paper tackles unsupervised learning scenarios as well as supervised learning. They evaluate the proposed SCP on permutated MNIST, sequential learning in autoencoder task, and sequential learning for segmentation.

[Pros]
- This paper tackles unsupervised learning scenarios beyond the classification on benchmark datasets.
- This paper employes sliced Cramer distance with theoretical justification.
- The analysis on EwC and MAS in terms of geometric view
- Experimental results look promising.

[Cons]
- Even if MAS[1] describes the details on synaptic concept and Hebbian rule, many readers might be not familiar with synaptic or neuro-science terms. So, in prelimiary session, more explanation can help readers to understand.
- In addition to permute-MNIST, it is required to be evaluated on more conventional tasks such as MNIST->SVHN or CIFAR-10, 100 datasets. Also, more recent work should be compared such as IMM [2] and PGMA [3] for supervised learning scenarios.
- All graph result figures can be improved for enhancing legibility. In Figure 6, In particular, the subtitle of Summer case confuses me.

[1] Aljundi et al. Memory Aware Synapses: Learning what (not) to forget, ECCV 2018.
[2] Lee et al. Overcoming Catastrophic Forgetting by Incremental Moment Matching, NIPS 2017.
[3] Hu et al. Overcoming Catastrophic Forgetting for Continual Learning via Model Adaptation, ICLR 2019.

**Experience Assessment:**

I have published one or two papers in this area.

**Review Assessment: Checking Correctness Of Derivations And Theory:**

I assessed the sensibility of the derivations and theory.

**Review Assessment: Checking Correctness Of Experiments:**

I assessed the sensibility of the experiments.

**Review Assessment: Thoroughness In Paper Reading:**

I read the paper at least twice and used my best judgement in assessing the paper.

---

> ### Author Response · Authors · 2019-11-15
> **Response to Official Blind Review #3**
>
> Thank you for the positive evaluation of our work and your insightful comments. We appreciate your valuable time spent on serving the community. Below please find our responses.
>
> We agree with the reviewer that the neuro-scientific terms used in our paper might make our manuscript, unnecessarily, hard to follow for many readers. To that end, we have revised the Introduction and Preliminaries sections and added rigorous definitions of these terms to provide a more enjoyable read for the readers.
>
> We agree with both reviewers that the permuted-MNIST task is artificial and overly simple, which was precisely the reason for us to include the results on the SYNTHIA dataset in the paper. We also agree with the reviewer that our article could benefit from additional experiments; hence, we performed the requested experiments on MNIST-to-SVHN and sequential learning on CIFAR-100. Please see the extended supplementary materials in the revised manuscript.
>
> Regarding the comparison with IMM and PGMA, we first note that we have added these methods to our references. Next, we would like to point out the fundamental differences between the methods included in our work (i.e., online-EWC, MAS, and SCP) and IMM and PGMA. Below we enumerate these differences:
>
> The core idea behind IMM [2] is to learn a new model for the new task using the old optimized parameters as an initialization and then consolidate the old and new models into a single model via moment matching (mean-IMM or mode-IMM). However, in our case, we only utilize one model and use the concept of synaptic plasticity, where we selectively rigidify the network parameters to preserve the previously learned representation while learning a new task. As you can see, there is a fundamental difference between the two approaches, which makes the direct comparison challenging.
>
> PGMA [3] is yet another fascinating approach that is fundamentally different from, and arguably significantly more complicated than our proposed method. At its core, PGMA uses the concept of memory replay, which is an orthogonal approach to selective synaptic plasticity that is the focus of our work. In short, PGMA learns an autoencoder-based generative model for old tasks. To avoid catastrophic forgetting, PGMA then replays samples from old tasks using its generative model while learning the new task.
>
> Given: 1) the significant differences between IMM and PGMA and our proposed method, 2) the short period for the response, and 3) the unfortunate overlap of the ICLR rebuttal deadline and the CVPR2020 submission deadline, we could not provide the comparison to these methods at this time.
>
> Finally, we have updated the caption of Figure 6 to make it easier to follow. In particular, we clarify that the blue and red shadings on the plots indicate the durations in which the models were trained on ``summer'' and ``winter'' data, respectively.
>
> [1] Aljundi et al. Memory Aware Synapses: Learning what (not) to forget, ECCV 2018.
>
> [2] Lee et al. Overcoming Catastrophic Forgetting by Incremental Moment Matching, NIPS 2017.
>
> [3] Hu et al. Overcoming Catastrophic Forgetting for Continual Learning via Model Adaptation, ICLR 2019.

---

### Decision · Program_Chairs · 2019-12-19

**Decision:**

Accept (Spotlight)

**Comment:**

The paper addresses an important problem (preventing catastrophic forgetting in continual learning) through a novel approach based on the sliced Kramer distance. The paper provides a novel and interesting conceptual contribution and is well written. Experiments could have been more extensive but this is very nice work and deserves publication.